# Protein Supplementation with Short Peptides Prevents Early Muscle Mass Loss after Roux-en-Y-Gastric Bypass

**DOI:** 10.3390/nu14235095

**Published:** 2022-12-01

**Authors:** Marta Comas Martínez, Enzamaria Fidilio Meli, Fiorella Palmas Candia, Efrain Cordero, Irene Hernández, Ramon Vilallonga, Rosa Burgos, Anna Vila, Rafael Simó, Andreea Ciudin

**Affiliations:** 1Endocrinology and Nutrition Department, Hospital Universitari Vall d’Hebron, Diabetes and Metabolism Research Unit, Vall d’Hebron Institut de Recerca (VHIR), Universitat Autonoma de Barcelona, 08035 Barcelona, Spain; 2Research Group M3O, Methodology, Methods, Models and Outcomes of Health and Social Sciences, Faculty of Health Sciences and Welfare, University of Vic–Central University of Catalonia, 08500 Vic, Spain; 3Endocrine, Metabolic and Bariatric Unit, General Surgery Department, Hospital Universitari Vall d’Hebron, Universitat Autonoma de Barcelona, 08035 Barcelona, Spain; 4Centro de Investigacion Biomedica en Red de Diabetes y Enfermedades Metabolicas Asociadas (CIBERDEM), Instituto de Salud Carlos III (ISCIII), 28222 Madrid, Spain

**Keywords:** protein, nutrition supplementation, fat-free mass, bariatric surgery, short peptides, complex protein, HMB-enriched formulas

## Abstract

Introduction: A significant reduction in fat-free mass (FFM) following bariatric surgery (BS) has been reported, and adequate protein intake is recommended for FFM preservation. Current guidelines of nutritional management after BS recommend complex protein (CP) compounds. However, Roux-en-Y-gastric bypass (RYGB) has a negative impact on CP digestion, leading to protein malabsorption. At present, there is no data regarding the impact of early supplementation with short peptide-based (SPB) or hydroxy methylbutyrate (HMB)-enriched formulas on the evolution of the FFM after the BS. Aim: The aim of this study is to evaluate the effect of nutritional products based on CP, HBM-enriched, or SPB formulas on the FFM of patients that undergo RYGB. Material and methods: This is a prospective interventional study, including three groups of patients (according to the type of protein product) as candidates for BS, recruited between December 2021 and April 2022, matched by age, gender, and BMI. All patients underwent evaluations at baseline and one month post-BS, including: medical history, physical and anthropometric evaluation, bioimpedance, and biochemical analysis. Results: A total of 60 patients were recruited: 63% women, mean age 43.13 ± 9.4 years, and BMI 43.57 ± 4.1 kg/m^2^. The % of FFM loss from total weight loss (TWL) was significantly lower in the SPB group than CP and HMB groups despite the major %TWL in this group (40.60 ± 17.27 in CP, 34.57 ± 13.15 in HMB, and 19.14 ± 9.38 in SPB, *p* < 0.001). TWL% was 9.98 ± 1.82 vs. 9.83 ± 2.71 vs. 13.56 ± 4.30, *p* < 0.001, respectively. Conclusion: In our study, the SPB supplementation prevented almost 50% FFM lost from the TWL than the CP- or HMB-enriched compounds at one month post-BS. These results are significant in the setting of muscle mass preservation after the BS, and have the potential to change the current guidelines for the management of nutritional supplementation after BS.

## 1. Introduction

Bariatric surgery (BS) is currently the most successful treatment for severe obesity in terms of significant and sustainable weight loss, leading to substantial improvement in obesity comorbidities and quality of life [1]. However, several complications have been reported, some of which are related to malabsorption [2]. In this regard, Varus et al. [3] and Maimoun et al. [4] showed that a significant reduction in fat-free mass (FFM) occurs after the BS. Additionally, our group has recently showed that an early and significant loss of FFM occurs as early as 1 month after the BS and continues after 24 months, regardless of the BS technique (Roux-en-Y-gastric bypass (RYGB) or sleeve gastrectomy (SG)). This finding is even more important in the context of the recently described sarcopenic obesity [5], (low muscle mass and obesity) and has been related to metabolic complications, such as type 2 diabetes, sleep apnea, and unfavorable outcomes after the BS [6,7,8].

Current guidelines recommend daily physical exercise and adequate protein intake for muscle mass preservation [9,10,11]. However, emerging evidence suggests that there is a decrease in protein absorption after BS (in both RYGB and SG) due to particular changes in the anatomy and function of the gastrointestinal tract, as well as due to reductions in the volume of meals [12,13,14]. Existing guidelines of nutritional management after BS recommend supplementation with 60–80 g of protein per day or 1–1.5 g/kg of ideal body weight per day [15,16,17], starting from the first 48 h, especially during the first month after the BS.

Usually, the dietary proteins are complex, and 50% are absorbed in the duodenum and the rest through the small intestine. The RYGB malabsorptive component is based mainly on the duodenal exclusion. Additionally, the gastric acid pH and pepsinogen synthesis (determinants for the digestion of complex proteins) are significantly decreased after both RYGB and SG [12,13,14]. Furthermore, in many cases, the patients limit the ingestion of meat (the main source of proteins), especially in the first months after the BS [18]. All these factors can have a negative impact on the digestion and absorption of complex protein products after the BS, and consequently on the muscle mass.

It should be noted that the supplemental products recommended after the BS are based on complex proteins. Furthermore, in the daily clinical practice it is quite common that the high quantity of proteins recommended is not well-tolerated, and the patients are not compliant.

At present, different nutritional products with different types of presentations of proteins are available: complex proteins, caseinate-based proteins, whey-based proteins, peptide-based formulas, essential amino acids, and leucine precursors such as β-hydroxymethyl β-butyrate (HMB), among others [19].

Nevertheless, as far as we know, there is no reliable data regarding the impact of the early supplementation with these nutritional products on the evolution of the muscle mass after the BS.

Based on the above, we performed the present study aimed at prospectively evaluating the impact of different nutritional products based on complex proteins, enrichment with HBM, or short peptides-based compounds on the muscle mass of patients who underwent RYGB. Our hypothesis was that using nutritional products enriched with HBM or short peptides-based compounds will help to better preserve the muscle mass after the BS.

## 2. Materials and Methods

This was a prospective interventional study, including 60 patients with severe obesity as candidates for BS, recruited between December 2020 and April 2021. Patients were divided into 3 groups according to the protein supplementation products that were prescribed (see below) and matched by age, gender, and BMI prior to BS. The study was approved by the local Ethics Committee PR(AG)690/2020 and was carried out following the Declaration of Helsinki. All the patients signed the informed consent form before inclusion in the study. To obtain a more homogeneous sample, only patients who were candidates for RYGB were selected.

Inclusion criteria: (a) Age between 18 and 60 years, (b) BS criteria as per the protocol (BMI > 40 kg/m^2^ regardless of comorbidities, or BMI > 35 kg/m^2^ with at least one obesity-related comorbidity), (c) completion of the preoperative protocol for BS at our site, and (d) signed a written informed consent form.

Exclusion criteria: (a) Patients undergoing evaluation for second-stage surgery, (b) unable to perform post-BS follow-up at our center, (c) unable to perform BIA (e.g., limb amputation, unwillingness, and unable to fast for more than 8 h), (d) presence of other conditions that may affect muscle mass according to investigators’ criteria (e.g., immobilization, myopathies, and endocrinopathy such as Cushing’s disease), (e) severe concomitant pathology (cardiovascular, cerebrovascular, pulmonary, renal, or neoplastic) which may limit study participation according to the investigators’ criteria, (f) use of drugs that can affect muscle mass (e.g., corticosteroids), (g) active drug or alcohol abuse, (h) uncontrolled psychiatric illnesses or eating disorders, and (i) type 2 or type 1 diabetes.

At baseline and at one month after the BS, all patients underwent the following: complete medical history, physical and anthropometric evaluation, bioelectrical impedance analysis (BIA), and biochemical analysis comprising sensitive parameters of protein metabolism (transthyretin), as per the preoperative BS protocol at our site.

BMI was calculated using the formula: weight (kg)/height^2^ (m^2^) [20].% Total weight loss (% TWL) was calculated using the formula: (Initial W (kg) − Final W (kg))/(Initial W (kg)) × 100.FFM to TWL was calculated using the formula: (Initial FFM (kg) − Final FFM (kg))/(Initial W (kg) − Final W (kg)) × 100 [2,3].

According to our site’s postoperative protocol, patients received an exclusively liquid diet with nutritional supplements (both shake and protein powder) during the first two weeks, with a total of 80–90 g of protein per day or 1.5–2 g/kg ideal body weight/day. If there was good tolerance, in the next 15 days, a progression to crushed texture was started combined with the nutritional supplements. A month after the intervention, a progression to an easy-to-chew and easy-to-digest diet combined with the support of protein powder was recommended. Depending on the patient’s tolerance, in the following months, more complex food was introduced until a complete balanced diet was acquired. During this process, the patients continued taking protein powder supplementation to achieve protein requirements.

The nutritional products used in the study are reflected in Table 1.

Control product: specific nutritional product for patients who underwent surgery and required hypocaloric diets (calorie restriction).Product enriched with HMB (β-hydroxymethyl β-butyrate).Peptide-based formula.

The compliance was evaluated using a specific questionnaire designed for this study (see Appendix A). In addition, physical exercise was prescribed according to the current guidelines [17,21,22]. 

The surgical technique used was performed by the same team, formed by three trained bariatric surgeons, as the standard Roux-en-Y-gastric bypass (RYGB) [23], and had the following characteristics: food loop length: 150–180 cm, biliopancreatic loop length: 120 cm, gastric pouch: 30 m^3^.

Body muscle mass was assessed by multi-frequency bioelectrical impedance analysis (BIA) [24]. The BIA device used was the Bodystat QuadScan4000^®^. For the body composition measurement, the surface electrodes must be placed on the right side of the body. The electrodes are placed on the metacarpal line and the metatarsal line of the hand and foot on the same side of the body. The measurement can be performed in the outpatient clinic and hospitalization areas since this device is portable [25]. The patients included in the study were asked to meet the standardized conditions to perform the BIA: avoid physical exercise for the previous 8 h, fast for 6–8 h before the measurement, including water, remove all metal objects that may interfere with the measure, and if the patient wears a prosthesis or implant, the measurement was performed on the opposite side [26].

The variables collected from the BIA evaluation were fat mass (FM) (kg), fat-free mass (FFM) (kg), fat-free mass index (FFMI) (kg/m^2^), body cell mass (BCM) (kg), resistance (R) (Ω), reactance (Xc) (Ω), impedance (Z) (Ω), and phase angle (PA) (°). PAs are considered an indicator of cell integrity, and this allows the interpretation of the capacity of the cell to transmit the electrical stimulus produced by the BIA device.

### Statistical Analysis

IBM SPSS version 25 software was used. Continuous variables are expressed as means ± standard deviation (SD) for normally distributed variables and median ± interquartile range (IQR) for non-normally distributed variables. Categorical variables are expressed as percentages. ANOVA and Student’s t-test were used for differences between groups of variables. Spearman’s correlation and logistic regression analyses were used to assess the relationship between different variables. A *p*-value < 0.05 was considered statistically significant.

## 3. Results

A total of 80 patients were recruited. All the patients underwent preoperative evaluation for BS at our center as well as a Roux-en-Y-gastric bypass. Only patients that presented a compliance with the nutritional products of ≥75% were evaluated after one month. A total of 20 patients were excluded due to this reason (data shown in Appendix A).

The baseline characteristics of the patients are shown in Table 2.

Regarding the impact of the different nutritional products on the FFM and body composition parameters after one month post-BS, the data are shown in Table 3.

FFM loss from the TWL was significantly lower with the product containing short peptides in comparison with the other two products as shown in Table 4 (*p* < 0.05).

The organoleptic evaluation of the nutritional products is displayed in Table 5. We observed a general good acceptance of the supplementation, but the product with the best score in all organoleptic components was the control product.

Side effects were registered in 27% of the patients, and the most common were nausea (10%) and vomiting (8%). Of the patients, 32% commented that they wanted a greater variety of flavors, 13% reported that they tasted too sweet, and 12% explained that they found the texture too thick.

## 4. Discussion

As far as we know, this is the first prospective study showing that supplementation with short peptides after the BS prevents significant muscle mass loss, in comparison with complex proteins and HMB-enriched formulas. These findings are important in the actual context when evidence is accumulating that BS has a negative and early impact on muscle mass [2,3], and protein supplementation is almost the exclusive source during the first month [15,16,17]. In this regard, our group has recently shown that the pre-BS muscle mass was an independent determinant of muscle loss after the BS [2]. This finding is relevant because sarcopenic obesity is related to unfavorable outcomes after the BS [6,7,8].

The specific changes induced by the BS, and in particular RYGB, have a negative impact on the digestion and absorption of complex proteins. Despite the evidence, at present there is no specific recommendation regarding the protein products that should be used after the BS. The Enhanced Recovery After Surgery (ERAS) guidelines for BS recommend a standard intake of protein and carbohydrates, using an iso-osmolar drink 2–3 h before surgery to avoid loss of protein and muscle mass. Postoperative nutrition is not more specific: they recommend protein intake between 60 and 120 g per day without specifying the type or format of the protein [27]. Additionally, the European Society for Clinical Nutrition (ESPEN) nutritional guidelines recommend 60 g of protein/day after BS to minimize the loss of fat-free mass, but without specifying the type of protein that should be used [28].

Proteins can be differentiated by their absorption and digestion: fast (made of whey) or slow (made of caseinate). Some studies concluded that whey-based protein helps to preserve muscle mass [29,30]. Gilmartin et al. [31] showed that specific essential amino acids intake, especially leucine, improves muscle mass, together with whey protein and exercise. Furthermore, recent data indicated that β-hydroxymethyl β-butyrate (HMB), a precursor or metabolite of leucine, has anabolic effects on protein metabolism [32,33], and suggested that it could be a valid option for optimal protein supplementation. The recommended amount of HMB (3 g/day) is difficult to obtain through a regular diet, and therefore, external supplementation is necessary. However, these studies were performed in an elderly population, and at present there is no data on patients undergoing BS.

In our study, we compared the impact of complex protein intake (classical supplementation), HMB-enriched products, and short peptides protein on the muscle mass at one month after the RYGB. It should be noted that only data from those patients that had a compliance of >75% with the nutritional protein supplementation product were analyzed. No statistical differences were seen between the complex protein and the HMB-enriched product regarding the percentage of FFM lost from TWL, suggesting that despite the HMB enrichment, this product still contains a complex protein, whose absorption and metabolism can be negatively impacted by the RYGB. In contrast, patients who received the product containing short peptides lost almost 50% less of FFM in comparison with the other two groups. In the case of a nutritional product based on hydrolyzed protein, the processes of digestion and absorption do not require the complex enzymatic degradation, and duodenum and can be easily absorbed from the jejunum [34,35]. Some experimental evidence has shown that intestinal adaptation can occur after RYGB, thus resulting in an improvement of the digestion/absorption of complex proteins [36]. In addition, a study in humans (*n* = 9) did not show any significant alteration in protein digestion three months after RYGB [37]. However, data in humans are limited, and it seems unlikely that intestinal adaptation occurs as early as one month after the BS.

Additionally, since the recommended amount of protein intake is high compared to a standard diet, the compliance of the patients should be carefully evaluated in daily clinical practice. The gastrointestinal tolerance of protein supplementation is a major concern. Recent data showed that peptide products are better-tolerated than complex protein [38], while other studies showed no differences in this regard [39]. In our study, 25% of the patients had a lower compliance, but no differences were seen between the three products in terms of tolerability and acceptance.

The main limitations of our study might be the use of bioimpedance as a method of evaluating the body composition and the short period of follow-up. Nevertheless, despite the errors that were described regarding the use of bioimpedance, they are not significant if this method is used for the follow-up of the same patient, as is the case of our study. Additionally, we chose to evaluate the evolution of the muscle mass as soon as one month after the BS because this timepoint is when most of the muscle mass is lost, and an early adequate nutritional intervention will help prevent the muscle mass loss.

## 5. Conclusions

We have provided the first evidence that that hydrolyzed protein supplementation lost almost 50% less FFM from the TWL than complex protein or HMB-enriched compounds one month after BS. This finding is crucial because the preservation of the muscle mass as early as in the first weeks after the BS is an independent determinant of muscle mass evolution after the BS and overall morbimortality [2]. In this regard, our results could change the current guidelines for the management of nutritional supplementation after the BS. However, further studies aimed at confirming our results, as well as to explore the underlying mechanisms, are needed.

## Figures and Tables

**Table 1 nutrients-14-05095-t001:** Nutritional composition of the products used.

Nutrients	Control Product (50 g with 200 mL of Water)	Short Peptide Product (200 mL)	Product with HMB (220 mL)
Energy (kcal)	210	300	330
Protein (g)	15	13.5	20
Carbohydrates (g)	27.4	36.8	37
Fat (g)	4.5	11	11
HMB (g)	-	-	1.5
Carnitine (mg)	15	30	40
Choline (mg)	-	136	154
Taurine (mg)	15	30	-
Arginine (g)	15	-	-
Fiber (g)	-	-	-
Sodium (mg)	200	338	330
Potassium (mg)	620	400	594
Chlorine (mg)	320	300	139
Calcium (mg)	333	200	499
Phosphorus (mg)	168	200	260
Magnesium (mg)	52.5	60	55
Iron (mg)	4.2	4	4.6
Manganese (mg)	0.67	1	0.99
Copper (mcg)	450	480	539
Zinc (mg)	3.2	3.6	3.9
Iodine (mcg)	70	30	48
Selenium (mcg)	18.5	19	20
Chrome (mcg)	27.5	16	19
Molybdenum (mcg)	28	36	33
Fluorine (mg)	0.3	-	-
Vitamin A (mcg)	305	300	264
Vitamin D (mcg)	1.8	2	13
Vitamin E (mg)	5	3.8	5.5
Vitamin C (mg)	27	36	35
Vitamin K (mcg)	25	14	33
Folic acid (mcg)	105	60	77
Vitamin B1 (mg)	0.5	0.42	0.57
Vitamin B2 (mg)	0.6	0.60	0.70
Vitamin B6 (mg)	0.6	0.60	0.66
Vitamin B12 (mcg)	0.5	1	1.4
Niacin (mg)	5.5	6	6.6
Pantothenic acid (mg)	2.3	2	2.4
Biotin (mcg)	15.8	11	1.3
Lactose (g)	5	-	-
Fructose (g)	4.1	-	-

**Table 2 nutrients-14-05095-t002:** The baseline characteristics of the patients included in the study.

*n*	60
Gender (women %)	38 (63%)
Age (years) mean ± SD	43.13 ± 9.4
BMI before BS (kg/m^2^) mean ± SD	43.57 ± 4.11

BMI: body mass index, SD: standard deviation.

**Table 3 nutrients-14-05095-t003:** Body composition evolution before and 1 month after bariatric surgery according to nutritional products.

Parameters	Control Product		Product with HMB		Short Peptides Product	
	Baseline	1 Month after BS	Baseline	1 Month after BS	Baseline	1 Month after BS
Fat mass (FM) (kg)	50.80 ± 7.33	43.66 ± 7.53	50.97 ± 10.28	44.08 ± 10.48	58.25 ± 6.83	45.12 ± 7.46
Fat-free mass (FFM) (kg)	69.76 ± 10.10	64.89 ± 9.28 *^b^*	58.20 ± 12.72	54.27 ± 11.73 *^b^*	62.81 ± 11.54	59.28 ± 10.31
Body cell mass (BCM) (kg)	44.98 ± 5.44	40.33 ± 4.96 *^b^*	38.96 ± 7.61	35.22 ± 6.58 *^b^*	41.81 ± 7.03	37.15 ± 5.74
BMI (kg/m^2^)	42.51 ± 3.56	38.28 ± 3.23	43.45 ± 4.37	39.19 ± 4.25	44.77 ± 4.47	38.62 ± 3.43
FFMI (kg/m^2^)	24.43 ± 1.76	22.76 ± 1.66 *^a^*	22.93 ± 2.37	21.38 ± 2.29	23.08 ± 3.15	21.78 ± 2.71 *^a^*
Resistance (R) (50 kHz) (Ω)	402.60 ± 37.10	457.2 ± 51.19	409.40 ± 45.49	454.7 ± 52.38	424.30 ± 70.56	452.20 ± 71.11
Reactance (Xc) (50 kHz) (Ω)	49.50 ± 8.66	52.32 ± 7.90 *^a^*	44.80 ± 7.16	48.93 ± 13.10	48.00 ± 6.13	46.94 ± 5.49 *^a^*
Impedance (Z) (50 kHz) (Ω)	405.70 ± 37.62	460.2 ± 51.68	410.80 ± 45.48	457.4 ± 53.02	427.00 ± 70.53	454.90 ± 70.91
Phase angle (PA) (º)	7.03 ± 0.99	6.54 ± 0.75	6.29 ± 0.96	6.10 ± 1.25 *^b^*	6.55 ± 0.96	6.01 ± 0.99 *^a^*
CK (UI/l)	126.90 ± 35.04	92.20 ± 55.18	127.90 ± 71.15	88.90 ± 63.78	131.60 ± 97.42	76.70 ± 59.74 *^a^*
Protein (g/dL)	7.10 ± 0.46	6.66 ± 2.39	7.25 ± 0.33	7.15 ± 0.51	7.10 ± 0.49	7.04 ± 0.57
Albumin (g/dL)	4.21 ± 0.030	4.02 ± 1.47	4.18 ± 0.30	4.35 ± 0.39	4.22 ± 0.29	4.25 ± 0.31
Transthyretin (mg/dL)	24.47 ± 3.90	16.39 ± 11.39	25.42 ± 5.47	21.86 ± 5.16 *^b^*	24.26 ± 2.34	18.59 ± 3.75 *^a^*

BS, bariatric surgery; FFMI, fat-free mass index; CK, creatine kinase; BMI, body mass index. *^a^* Significantly different between the control product and the short peptides product at *p* < 0.05. *^b^* Significantly different between the control product and the product with HMB at *p* < 0.05.

**Table 4 nutrients-14-05095-t004:** Comparison between different nutritional products.

Parameters	Control Product	Product with HMB	Short Peptides Product
FFM from TWL (%)	40.60 ± 17.27 ^*a*^	34.57 ± 13.15 ^*c*^	19.14 ± 9.38 ^*a*,*c*^
TWL (%)	9.98 ± 1.82 ^*a*^	9.83 ± 2.71 ^*c*^	13.56 ± 4.30 ^*a*,*c*^
FFM loss (kg)	4,87 ± 2.39	3.93 ± 2.32	3.53 ± 2.81

FFM, fat-free mass; TWL, total weight loss; HMB, β-hydroxymethyl β-butyrate. *^a^* Significantly different with respect to the control product and the short peptides product at *p* < 0.05. *^c^* Significantly different with respect to the short peptides product and the product with HMB at *p* < 0.05.

**Table 5 nutrients-14-05095-t005:** Organoleptic evaluation of the protein supplements used in the study.

Organoleptic Values	Good Acceptance		
	Control Product 30% (24)	Product with HMB 35% (28)	Short Peptides Product 35% (28)
Flavor/taste	91.7% (22)	64.3% (18)	71.4% (20)
Smell	91.7% (22)	78.6% (22)	71.4% (20)
Color	91.7% (22)	85.7% (24)	85.7% (24)
Tolerance	83.3% (20)	71.4% (20)	71.4% (20)

## Data Availability

Data are available upon direct request to the corresponding author.

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
