# Peer review of "Protein Supplementation with Short Peptides Prevents Early Muscle Mass Loss after Roux-en-Y-Gastric Bypass"

_nutrients, 2022, doi:10.3390/nu14235095_

Round 1
Reviewer 1 Report
General comments
This manuscript aims at prospectively evaluating the impact of different nutritional products based on complex proteins, enrichment with β-hydroxymethyl β-butyrate or short peptides-based compounds on the muscle mass of patients underwent to Roux-en-Y-gastric bypass. Despite some minor issues detailed below, authors manage to fulfill sufficiently their aims.
Minor comments
(line 29 and elsewhere throughout MS) Please, do not start sentences with acronyms; … m2… (viz., “2” subscript. This elsewhere throughout MS, as well);
(l31) please, be consistent regarding “.” or “,” use as decimal separator;
(l37) please, use “;” to separate different keywords each other;
(l43 and elsewhere throughout MS, I refer to “.” placement) … life (1). However…
(l271) 2. Martínez…
(l48) … (Roux-en-Y-gastric bypass (RYGB) or sleeve gastrectomy (SG));
(l108) … evaluation, bioelectrical impedance…
(l112÷5) please, check for font size;
(l133) within submission there isn’t any “Supplementary material”;
(l138) “mL3”?
(l170) … different nutritional products…
(l203) please, introduce ERAS;
(l207) please, introduce ESPEN;
(l347) extra line;
(l257) … morbimortality (2);
(l260) please, add authors’ contributions;
(l264÷7) please, add Institutional Review Board Statement, Informed Consent Statement, Data Availability Statement and Acknowledgments.
Author Response
Answer: First of all we would like to thank Reviewer 1 for the comments and constructive criticism of the study. All the recommendations have considerably increased the quality of our manuscript. We have corrected all the typos.
Issue (l133) within submission there isn’t any “Supplementary material”;
Answer: Thank you very much for the observation. There was an error of submission, we missed these files. We have included the supplementary material in the system: the questionnaire and the data from the patients that were excluded. Please note, that the tendency of losing less FFM in the group treated with short peptides is also seen in these patients that were less compliant with the supplementary products. Some of them were about 50% compliant. Nevertheless, since we can not asure a homogeneous compliance in these patients, we decided not to include them in the main study.
Yours sincerely,
Andreea Ciudin and Ramon Vilallonga
Reviewer 2 Report
The authors did an excellent job with the study and show evidence for that short-peptide supplementation reduces the Free-Fat-Mass loss from Total weight loss. Despite the sample number looks small and also has some exclusions, the data points out that this hydrolyzed protein supplementation could rescue the patients. These results help design better guidelines for nutritional supplementation for patients undergone Roux-en-Y-gastric by-pass.
Besides this important study outcome, data form at least 20 patients was excluded and attached in supplementary data(line 133, 166) as per authors. But I couldn’t find how to access the supplementary material, please let me have access to that information.
Author Response
First of all we would like to thank Reviewer 2 for the comments regarding our study. Regarding the supplementary material, thank you very much for the observation. There was an error of submission, we missed these files. We have included the supplementary material: the questionnaire and the data from the patients that were excluded. Please note, that the tendency of losing less FFM in the group treated with short peptides is also seen in these patients that were less compliant with the supplementary products. Some of them were about 50% compliant. Nevertheless, since we could not asure a homogeneous compliance within these patients (eben if lower) we decided not to include them in the main study.
Your sincerely,
Andreea Ciudin and Ramon Vilallonga